1.5em 0pt

# Amplitude Amplification for Quadratic Unconstrained Binary Optimization with Regression Based Neural Network Bootstrapping

Cody Kearse, Daniel Koch

*Abstract*—A series of recent studies has demonstrated that Quantum Amplitude Amplification (QAA), the generalization of Grover's search algorithm, is capable of solving combinatorial optimization problems using oracle operations which apply phases proportional to all possible solutions. However, the algorithm's success is highly sensitive to a free parameter choice which must be determined before running the quantum algorithm. In this study we demonstrate the feasibility of using regression neural network architectures to predict this parameter using only the weights and connections of a discrete objective function. We show that for both fixed length and varying length linear QUBO (quadratic unconstrained binary optimization) problems the neural network architectures can be trained to accurately predict the free parameter with sufficient error rates necessary for performing successful QAA.

*Index Terms*—Amplitude Amplification, Amp Amp, Quantum Computing, Graph Neural Network, Quantum Machine Learning, Hybridization, Hybrid Modeling, QUBO, Graph Convolutional Neural Network

## I. INTRODUCTION

Quantum Amplitude Amplification (QAA) is the generalization of Grover's search algorithm [1], and is a staple of quantum computational techniques. When marking a single or subset of states, Grover's algorithm has been proven to be optimal [2], [3], capable of reaching 100% probability of success [4]–[6], and has been experimentally demonstrated up to 5 qubits [7]–[10], [12]–[14]. Additionally, QAA itself has been incorporated as a subroutine in more complex quantum algorithms [15]–[19] that go beyond database searching.

In light of QAA's mathematical and theoretical success, physically realizing the core unitary operators of the algorithm has proven difficult due to circuit depth scaling [20], [21]. The work of Pokharel & Lidar [12] represents the extent to which QAA is achievable on today's hardware, incorporating state-of-the-art circuit optimization [22]–[24], dynamical decoupling [25], and quantum error detection [26]. All of these techniques are needed in order to achieve the $N$-qubit control operations necessary for both the oracle and diffusion operators of QAA. However, as discussed in Stoudenmire & Waintal's recent work [27], there is doubt as to whether an $N$-control unitary implementation of the oracle operator is computationally advantageous.

Address: Air Force Research Laboratory, Rome, NY 13441, USA.
Emails: cody.kearse@us.af.mil, & daniel.koch.13@us.af.mil

In this study we focus on a formalism of QAA that uses oracle operations which apply phases proportional to all possible solutions of a discrete cost function [28]–[30], which we refer to as *cost oracles* [31]–[33]. The variant of QAA discussed in this work has strong overlap with a variant of QAOA which utilizes *Grover-Mixers* (diffusion) [34]–[36] together with the *phase-separator* operator (cost oracle). Similar to Grover Adaptive Search [37] (GAS), the goal is to find the extrema min/max solution(s) of a weighted combinatorial optimization problem, in this case QUBO (quadratic unconstrained binary optimization). Unlike the oracle construction of GAS which requires the creation of quantum dictionaries [38], cost oracles are very circuit depth efficient [32], especially on hardware architectures with support parallel gate operations such as superconducting qubits [39]. It has been shown that QAA encoding QUBO problems as the oracle is able to achieve probabilities of $90\%+$ for solutions near the global optimum [28], [30]–[32], but its success is very sensitive to the choice of a single free parameter in the oracle operation. Methods for approximating an optimal value for this parameter have been proposed [30], [32], but have yet to demonstrate the required precision necessary for large problem sizes.

The results of this study contribute to the growing research literature of applying machine learning to enhance quantum algorithms [40]–[47]. Dawid et. al. highlight several broad categories in which machine learning has assisted in quantum research: reproducing Hilbert spaces with kernels [41], quantum dynamics and physics modeling [42], wave function representation [43], quantum-feedback control [44], optimization [45], error-correction [46], and circuit-parameterization [47]. Our methods in this study are similar to those of Swaddle [48], utilizing neural network architectures to assist QAA.

The novel contribution of this work is the demonstration that a machine learning based approach can predict the optimal value for this free parameter to a level of precision which enables reliable QAA success. In the work of [32], [33] it was observed that there is a strong mathematical correlation between the numerical value of cost function solutions and the optimal free parameter value for finding them via QAA. Deep neural networks in particular have been shown to approximate a wide range of mathematical structures and phenomena with remarkable precision [49]–[52]. Aa a modern example, physics-informed neural networks (PINNs), have a strong history of producing numerical solutions for higher order differential equations for fluids dynamics with Navier-Stokes Equations [53], weather forecasting [54], modeling

thermodynamic systems, and even the prediction of ground state Hamiltonian's for quantum many body problems [50], [55]. In [56], Cichy argues there are even specific use-cases in which the predictive power of neural networks can help to jump-start the beginning of a larger theoretical framework. Given the well documented success of neural networks to regressively model mathematical phenomena, here we demonstrate that a highly performative regression model with strong generalization can capture to a sufficient degree, the mathematical link between the weights of a discrete cost function and the optimal cost oracle parameter value for finding the global optimum.

## II. AMPLITUDE AMPLIFICATION

We begin with an overview of the form of Amplitude Amplification that we seek to improve in this study, given by Alg. 1 below.

---

**Algorithm 1** Amplitude Amplification Algorithm

---

Initialize: $|\Psi\rangle = |0\rangle^{\otimes N}$
Prepare: $H^{\otimes N}|\Psi\rangle = |s\rangle$
**for** $k$ iterations **do**
    Oracle: $U_\text{C}(p_\text{s})|\Psi\rangle$
    Diffusion: $U_\text{s}(\theta)|\Psi\rangle$
Measure $|\Psi\rangle$

---

The strategy of QAA begins with the preparation of $|s\rangle$, the equal superposition state given in equation 1. For the $N$-qubit QUBO problems presented in this study, $|s\rangle$ represents the state of equal uncertainty across all $2^N$ possible solutions.

$$|s\rangle = \frac{1}{\sqrt{2^N}} \sum_i^{2^N} |Z_i\rangle \tag{1}$$

Following the preparation of $|s\rangle$, the structure of QAA is to apply alternating oracle and diffusion operators, given in equations 2 and 3. After $k$ iterations of oracle and diffusion, the algorithm concludes with a measurement on every qubit in $|\Psi\rangle$, collapsing the quantum system down to single state $|Z_i\rangle$ in the computational basis ($|0\rangle$ and $|1\rangle$ for every qubit). If done successfully, the measured state $|Z_i\rangle$ is the desired solution to the QUBO cost function C(Z) encoded by the oracle $U_\text{C}$.

$$U_\text{C}(p_\text{s}) = \sum_j^{2^N} e^{i\text{C}(Z_j)\cdot p_\text{s}} |Z_j\rangle\langle Z_j| \tag{2}$$

$$U_\text{s}(\theta) = \mathbb{I} - (1 - e^{i\theta})|\text{s}\rangle\langle\text{s}| \tag{3}$$

Diffusion as defined in equation 3 above is controllable by the free parameter $\theta$, but in this study we only consider the case of $\theta = \pi$ [28]–[32] as the focus of this study is to use ML for determining optimal $p_\text{s}$.

### A. QUBO Cost Oracles

In this study we consider oracle operations according to equation 2 which apply phases proportional to all possible solutions of a cost function C(Z). Implementing $U_\text{C}(p_\text{s})$ as a

gate-based circuit is very depth efficient [32], [39] as compared to Grover's [12], [37], in some instances equivalent to the phase-separator operator in the QAOA literature [34]–[36]. Given in equation 4 below is the QUBO cost function C(Z) which will be the focus of this study, where $z_i \in \{0, 1\}$ are binary variables which are represented by the computational basis states $|0\rangle$ and $|1\rangle$ of each qubit within $|Z_i\rangle$.

$$\text{C}(Z_i) = \sum_i^N W_i z_i + \sum_{ij} w_{ij} z_i z_j \tag{4}$$

Implementing a quadratic cost function as a cost oracle $U_\text{C}$ requires only single and 2-qubit phases gates. Specifically, $P(\theta)$ on every qubit for the linear weights $W_i$, and likewise $CP(\theta)$ between qubits for every $w_{ij}$ between nodes, given in equation 5 below. We shall use the terms *node* and *edge* for referring to the weighted graph interpretation of QUBOs, where each qubit represents one node ($W_i$) and each edge is a pair of nodes that share a connection ($w_{ij}$).

$$\text{P}(\theta) = \begin{bmatrix} 1 & 0 \\ 0 & e^{i\theta} \end{bmatrix} \quad \text{CP}(\theta) = \begin{bmatrix} 1 & 0 & 0 & 0 \\ 0 & 1 & 0 & 0 \\ 0 & 0 & 1 & 0 \\ 0 & 0 & 0 & e^{i\theta} \end{bmatrix} \tag{5}$$

With $U_\text{s}(\pi)$ as the diffusion operator, the success of QAA according to Alg. 1 boils down to correctly determining $p_\text{s}$ and $k$, where $p_\text{s}$ stands for *phase scale* as its roles is to effectively scale all of the solutions of C(Z) to a range of approximately $2\pi$. For large $N$ problem sizes it has been shown that $k$ can be approximated to the familiar $\approx \frac{\pi}{4}\sqrt{2^N/M}$ of Grover's [33] (for $M$ marked states), which leaves $p_\text{s}$ as the sole parameter to be determined. This problem of determining $p_\text{s}$ can be understood as the phase matching condition in traditional Grover's [57], which for cost oracles exists for each possible solution. In section IV we show that machine learning can accurately predict optimal $p_\text{s}$ values for QUBOs composed of randomized weights $W_i$ and $w_{ij}$.

## III. MACHINE LEARNING ARCHITECTURES

The core of our approach is to formulate the determination of $p_\text{s}$ into a machine learning task, specifically neural network regression in the form of mathematical modeling. We show that a machine-learning regression algorithm can bootstrap the computational relationship between QUBO node values $W_i$, edges $w_{ij}$, and the predicted $p_\text{s}$ parameter. Our model uses upfront training costs in the form of simulating QAA using QUBO cost oracles to determine optimal $p_\text{s}$. After training our model produces $p_\text{s}$ values for newly generated QUBOs almost instantaneously, with only model instantiation and throughput speed as limitations.

### A. Regression Formulation

Much like other classical regression tools, machine-learning regression algorithms can produce accurate numerical approximation(s) given a set of starting feature values and their corresponding output values [58]. Typically through back-propagation, a network optimizes or trains to mimic a functional

mapping of the input and output feature space. In practice, regression neural networks are deployed in cases where the functional mapping is assumed to be challenging in both the dimensionality of the feature space and complexity of the nonlinear relationships [59]. In modern algorithms, the nonlinear activation functions of each neural node and the density of node connections allows for extremely expressive learning and modeling facilitation [52].

### B. Advanced Architectures

During the later stages of experimentation, the complexity of the QUBO structures increased such that a standard feed-forward neural network struggled to reach reasonable training thresholds. This warranted the investigation of more advanced machine learning architectures including Long-Term Short-Term (LSTM) recurrent networks and hybrid models. It is common in machine learning practice to scale models proportionally with the increase in task demands [59]–[61]

LSTM recurrent networks are capable of modeling data in sequential order to further analyze the relationships between each feature. For QUBO Analysis it was theorized that layers of LSTM cells could learn the relational nuances between each node and edge value. The feature space of nodes and edges is formatted linearly and consequently can be interpreted and fed into an a bidirectional LSTM layer without additional overhead.

Within an LSTM cell block the algorithm maintains a rudimentary form of memory within a hidden state that is passed along with each sequential feature step. Next, the Forget Gate determines the value of information within the hidden state, either concatenating the information for the forward pass with a value close to one, or diminishing it's additive effect with values close to zero. The newly updated candidate cell is is then passed through the output gate. This final output is both the output from the cell during this forward pass as well as the updated hidden state for the next sequential feature pass. [62]. A detailed breakdown of the Long-Term Short-Term cell workflow is presented below in figure one.

**LSTM Cell Equations:**

$$f_t = \sigma(W_f[h_{t-1}, x_t] + b_f) \quad \text{Forget gate} \quad (6)$$
$$i_t = \sigma(W_i[h_{t-1}, x_t] + b_i) \quad \text{Input gate} \quad (7)$$
$$\tilde{C}_t = \tanh(W_C[h_{t-1}, x_t] + b_C) \quad \text{Candidate state} \quad (8)$$
$$C_t = f_t C_{t-1} + i_t \tilde{C}_t \quad \text{Cell update} \quad (9)$$
$$o_t = \sigma(W_o[h_{t-1}, x_t] + b_o) \quad \text{Output gate} \quad (10)$$
$$h_t = o_t \tanh(C_t) \quad \text{Hidden state} \quad (11)$$

**Variable Definitions:**

| | |
|---|---|
| $x_t$ | Input at time step $t$ |
| $h_{t-1}$ | Hidden state from previous time step |
| $C_t$ | Cell state at time $t$ |
| $W_f, W_i, W_C, W_o$ | Weights for gates and candidate cell |
| $b_f, b_i, b_C, b_o$ | Bias vectors associated with each gate |
| $\sigma$ | Sigmoid activation function, $\dfrac{1}{1 + e^{-x}}$ |
| $\tanh$ | Hyperbolic tangent activation, $\dfrac{e^x - e^{-x}}{e^x + e^{-x}}$ |

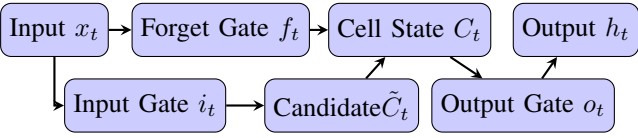

Fig. 1: LSTM Pipeline Overview

During algorithm development it also is a common practice to conjoin algorithm functions through hybridized layer types. Doing this can allow for richer numerical expression and combine the advantages of multiple algorithm techniques [63]. Common hybrid structures include LSTM or CNN input layers for feature extraction followed by dense linear layers for classification [64]. In this work, both LSTM layers and CNN backbone layers were conjoined with dense linear layers to support the increase in complexity of the research challenge for experiment 2.

## IV. METHODS

Predicting $p_s$ can be framed as a linear regression problem in classical machine learning. The values $W_i$ and $w_{ij}$ of the QUBO's cost function are used as input features of the network, and the output labels are the optimal $p_s$ values for finding global optimum (both min and max). In this section we demonstrate two cases where the mathematical structure underlying the connection between weights and $p_s$ is direct enough for the machine learning tools to predict $p_s$. The first use is a fixed length QUBO structure while the second varies the problem size. In the second case study we emphasize the improvements made to the ML architecture as compared to the first in order to address the challenges of scaling and problem flexibility for solving more complicated QUBO problems.

### A. Experiment 1: Fixed-Length QUBO

Let us now define the graphical representation of equation 4 which we shall use for the remainder of this study. Given a C(Z) composed of randomly generated integer weights from [−200,200] for both $W_i$ and $w_{ij}$, the corresponding graph is $n = N$ nodes with $m = N - 1$ connections between nodes in the manner shown in figure 2, which we define as a linear QUBO. Each node $n_i$ is assigned the linear weight $W_i$, and similarly each connection $m_i$ a quadratic weight $w_{ij}$. The objective is to predict the quantities $p_{\min}$ and $p_{\max}$ given the set of $n$ nodes and $m$ connections, which are the $p_s$ values which

maximize the probability of measuring the basis states $|Z_{\min}\rangle$ and $|Z_{\max}\rangle$ (the global optimal solutions to C(Z)) according to Alg. 1.

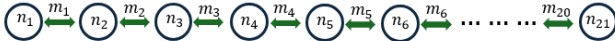

Fig. 2: Graphical representation of a length-$N$ linear QUBO.

There were special considerations for choosing both the length of the QUBOs $n$, as well as the node and edge values $n_i$ and $m_i$. First, $n = 22$ was chosen for experiment 1 due to the classical compute resources necessary for generating and solving QUBOs, ultimately warranting parallelization across multiple CPUs of our locally available high performance computing (HPC) systems. Data generation scales exponentially with the number of n nodes and m edges, creating an upper bound on the size of testable QUBO's, for which it was determined that $n = 22$ was the largest size at which a sufficient collection of data could be generated. It has been shown that as $n$ increases the overall performance and reliability of Alg. 1 improves [32], [33]. Thus, $n = 22$ is sufficiently large enough to produce results which represent the algorithm's large $N$ tendencies while also being reserved enough to generate the necessary training data within 2 weeks of compute time on a HPC cluster.

Second, the weights $n_i$ and $m_i$ were randomly selected to be integers from $[-200, 200]$ inclusively to guarantee good QAA performance [31]–[33], and to ensure that the training data would be representative of many possible problem instances. In total 100,000 QUBOs were generated in order to ensure our the training data would result in a model that is robust and generalizable over a wide variety of value settings. All data was normalized before training protocols within the neural network, which is a common practice in machine learning. The general symmetry of the distribution and normalization practices would also permit the model to be used tentatively with any data distributions that showed similar distribution qualities, further adding to its generalization capacity.

A traditional, fully-connected network of dense layers was determined to be the best choice for a preliminary training run. This architecture, while generic, would likely evidence any strong mathematical undertones within the problem space [49] [53]. It was reasoned that if a simple architecture could demonstrate promise, scaling and iterating through more complicated architectures would yield strong fine-tuned model performance [65]. Fine-tuning the preliminary model testing established the following architectural design and hyper-parameter optimization scheme for experiment one:

1. A standard feed-forward regression neural network with dense layers performed strongly for the preliminary QUBO task and was adopted as the architecture to fine-tune.
2. The width of the network was determined to be $n = 96$ for $m = 4$ hidden layers. Variations of $n = 32, 64, 128, 256$ were also considered as well as layer depths of $m = 3$ and 5.
3. The Mish activation function performed better than standard ReLU activations functions and was selected

for the hidden layers. This finding aligns with previous work utilizing the Mish activation function when modeling strongly represented underlying mathematical structures. [66]. The Mish function is defined as:

$$f(x) = x \tanh(\text{softplus}(x)), \tag{12}$$

$$\text{softplus}(x) = \ln(1 + e^x) \tag{13}$$

4. The learning rate was kept as default (0.001).
5. The LogCosh cost function was selected after showing better performance over standard mean squared error (MSE). This was attributed to the Gaussian nature of the problem space. The LogCosh function is defined as the following:

$$\text{LogCosh}(x) = \ln(\cosh(x)) \tag{14}$$

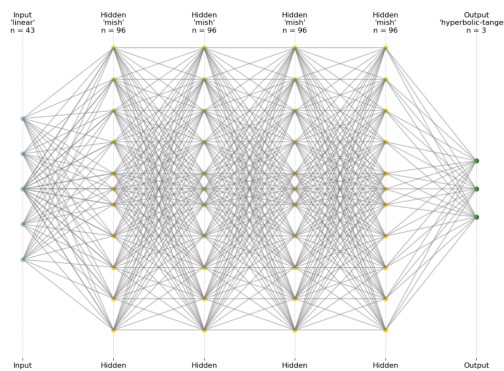

Fig. 3: Proof of Concept Network Configuration

### B. Experiment 2: Multi-length QUBO Generalization

For the second experiment we increased the complexity of the problem by creating a training set consisting of randomized length-$N$ QUBOs in addition to randomized weights. The data included QUBOs with node lengths between 18 and 23 inclusively. To improve upon the design of experiment 1 we created 100,000 samples of each length, bringing the total to 600,000 training examples (which includes the original $n = 22$ 100,000 from before).

To tackle the task of generalization within the range of QUBO sizes, more hyper-parameter tuning and architectural scaling was necessary for maximal performance. Additional alternative network configurations were tested which included a Long-Term Short-Term recurrent network (LSTM), a graph convolution neural networks (GCNN), transformer architectures, and hybridization combinations of differing network layers. During the trial processes graph convolution neural networks and transformer architectures were eventually eliminated from the testing criteria due to poor performance on the task. The best architectural choices and hyperparameters determined through fine tuning for the multi-length QUBO solver were as followed:

1. The best performing architecture was an LSTM backbone hybridized with dense layers connected at the front of the

pipeline. This aligned with our initial intuition to test the LSTM layers and noted that they performed strongly as the complexity of the solution space increased.

2. The best performing LSTM Backbone configuration included a single layer with 128 cells. Variations of the LSTM cell size including 32, 64, 256. We also tried differing the number of stacked LSTM layers of ( = 1, 2 ,3 ,4 ,5. The connected dense layer width of the network was determined to be $n = 512$ for $m = 3$ hidden layers. Variations of $n = 96, 128, 256, 1024$ were tested on hidden layer depths of $m = 1, 3, 5, 7$

3. Like the first experiment, the Mish activation function outperformed its ReLU counterpart at deeper training layers, where the number of epochs $N_s > 500$.

4. A learning rate of $0.01$ yielded better results than the default ($0.001$).

5. The LogCosh cost function was carried over into this experiment due to previous successes in experiment 1 and performed similarly.

## V. Results

Here we report on our models' ability to predict $p_s$ values for both experiments. For each model, 15% of the original data collection was isolated from training and kept as testing data, yielding a total of 15,000 and 90,000 QUBOs for the static and varied size experiments respectively.

### A. Metrics & Success Criteria

In the coming results we report on the percentage difference between our models' predictions of $p_{min}$ and $p_{max}$ versus their true values, given in equation 15. We track the average percent error as $\mu$ (the average of equation 15 across all 15,000 and 90,000 QUBOs respectively), and similarly the percent of all $p_s$ predictions that exceed an error of 2% as $\epsilon$. See figure 4 for an example which illustrates a gaussian profile centered at zero error.

$$\text{error} \% = \frac{|p_s - \text{prediction}|}{p_s} \qquad (15)$$

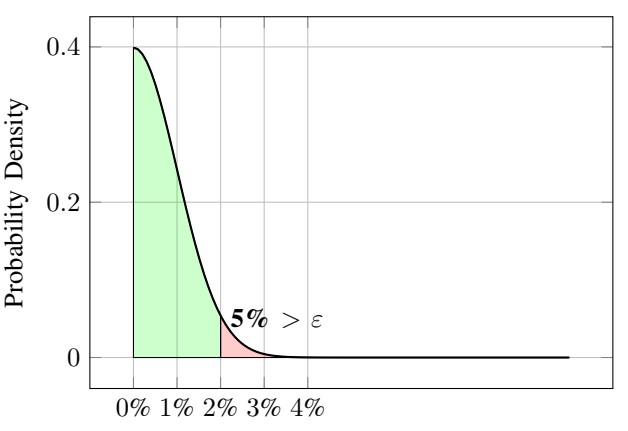

Fig. 4: Example distribution of $p_s$ predictions showing $\mu = 0.075\%$ and $\epsilon = 5\%$.

Based on the results of [28]–[33] as well as our own preliminary analysis on the QUBOs generated for experiments 1 and 2, it was determined that a 2% or less error in $p_s$ was needed to produce a computationally meaningful solution using QAA according to Alg. 1. To determine this threshold we analyzed the range of $p_s$ values that could produce $|Z_i\rangle$ solutions within the top 500 global optimum (see [32], [33] for more details) and found that 2% or less was necessary. In the Conclusions sections we discuss further the implications of these errors from using ML predicted $p_s$ values and potential QAA strategies as future research.

### B. Experiment 1: Static Length Linear QUBO's

An initial proof of concept experiment was able to successfully demonstrate that a neural network can learn relationships between QUBO node & edge weights and the optimal $p_s$ oracle parameters. Baseline model development was initiated with a standard series of three dense hidden layers in between a single input and output layer. Initial trials with this feedforward dense neural network produced regression prediction averages within 5-10% of the testing data values. Architectural adjustments in the number of layers, layer depth, and hyperparameter fine-tuning yielded improvements in the model such that the final version produced an average error of $\mu = 0.64\%$ and $0.68\%$ for $p_{min}$ and $p_{max}$ respectively on the testing data. The full results for this final model are shown in figure 5. This version of the model successfully predicted $p_s$ values under 2% error for a total of 96.5% of the 15,000 testing examples, resulting in $\epsilon \approx 3.5\%$. The distribution of errors closely matched the ideal distribution shown in figure 4, whereby the highest concentration of predictions were skewed towards 0% error followed by a sharp gaussian-like decrease away from zero.

### C. Experiment 2: Multi-length QUBO Generalization

The multi-length QUBO generalization of experiment 2 yielded particularly stronger results as compared to experiment 2, which may imply that the increased complexity of the training data gave the model more mathematical structure to learn. Initial model tests yielded regression predictions around an average of 2%, but after finetuning the $p_s$ predictions improved by nearly two orders of magnitude. The results of the final model are shown in figure 5, with mean errors of $\mu = 0.04\%$ and $0.05\%$ for $p_{min}$ and $p_{max}$ respectively. For this final version of the model only 108 instances of the 90,000 total test QUBOs produced errors above the 2% threshold, yielding $\epsilon = 0.12\%$. To achieve these results we note that the deep learning LSTM designs required 5 days of training time on a stand alone research laptop.

## VI. Conclusion

The results of section IV serve as a proof of principle that the $p_s$ problem of QAA for combinatorial optimization [30], [33] is solvable using classical machine learning techniques such as neural networks. In this study we demonstrated this functionality for linear QUBOs of both fixed and varying lengths, which suggests the same success may be possible for

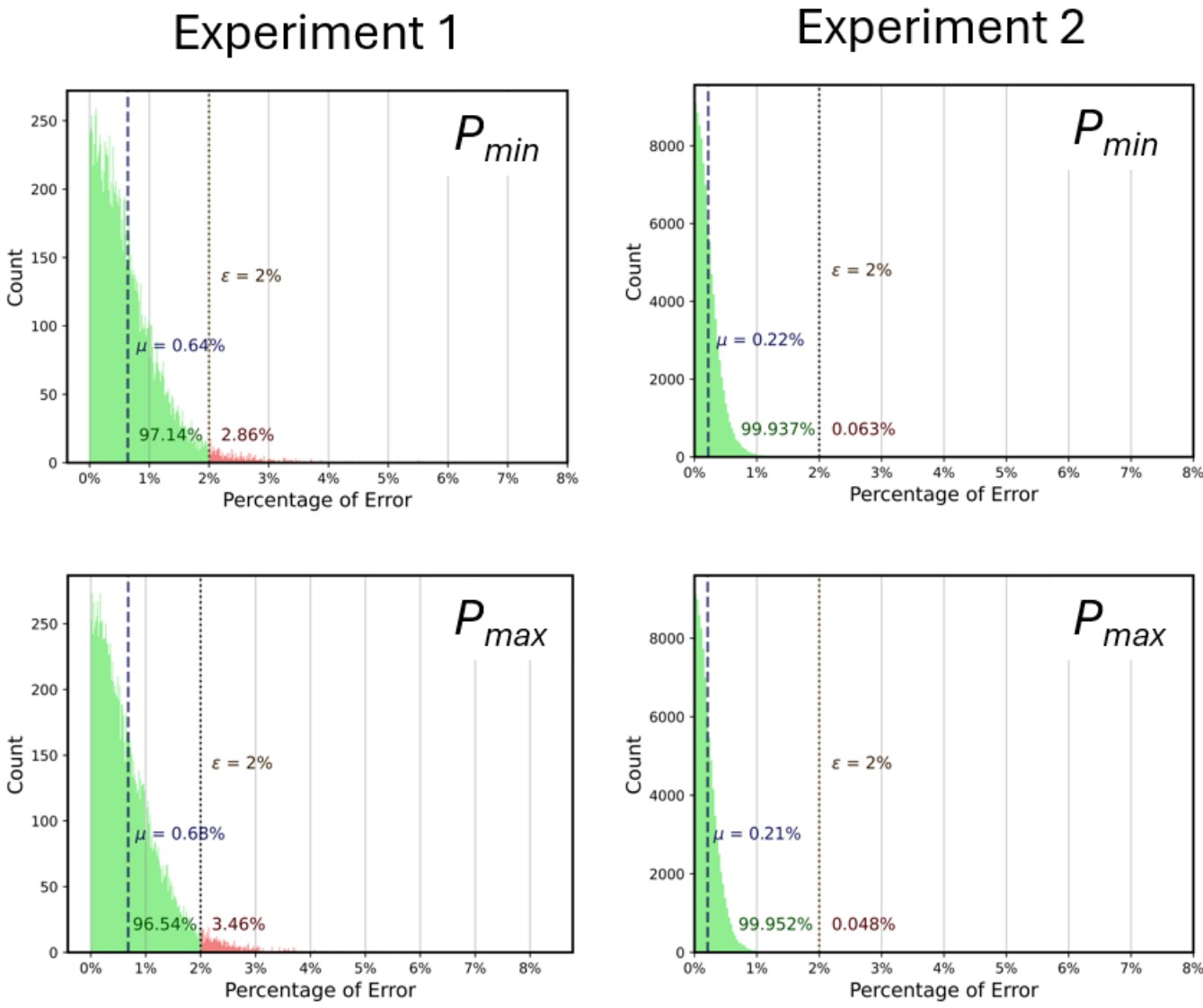

Fig. 5: Results from experiments 1 (left) and 2 (right) for our models' predictions on $p_{\min}$ (top) and $p_{\max}$ (right) on 15,000 and 90,000 QUBOs respectively.

larger and more complex problems. Because the mathematical relation demonstrated by our models was between only the weights of a cost function and $p_s$, this relation should in principle go beyond QUBO to HUPO (higher order unconstrained polynomial optimization) problems as well. These include use cases such as traveling salesmen, traffic modeling, supply chain, logistics and routing, where variables are not necessarily always binary (requiring qudits) and cost function weights go beyond quadratic. The ideal scenario for when classical ML solutions are applicable to assist Quantum Amplitude Amplification is one in which there is a high volume of continually new optimization problems that possess similar structure (ex. flight scheduling across airports or mail delivery routes across a city). Conversely, users in novel situations with rapidly evolving state-spaces might find significant challenges with integration [67]. Sufficient historical data of past problem instances is necessary for training, but after the upfront computational cost

of producing the models the payoff is in faster solutions to future problem instances.

### A. Future Work

Accuracy in predicting $p_s$ for QAA is step one towards a future hybrid computational pipeline, but more research is necessary to determine exactly how these approximate values can be utilized most effectively. The results of experiment 2 are particularly exciting, producing average errors around $0.05\%$ which exceeded our original expectations. Error within this order of magnitude or lower is accurate enough to deliver on globally optimal solutions, while conversely a $\mu \approx 0.5\%$ would warrant alternative QAA strategies to try and find near optimal solutions.

With regards to the machine learning side of the hybrid computation, there is still room for significant advances through architectural improvements. These include trimming

the algorithm to the smallest size possible while still delivering on high performance metrics, as well as testing differing learning protocols including potentially quantum variations like QNN's (quantum neural networks) and other quantum techniques. Testing these variations of learning algorithms will be crucial for scaling the technique to more complex optimization problems.

And finally, perhaps the most important consideration for the future viability of assisting QAA is to problem sizes beyond classically simulatable, which is the motivation for quantum computing to begin with. Neural networks are excellent at interpolation within the range the are trained on, but have limited use when generalizing outside of their training range. As we consider our hybrid approach for future applications it is important to remember that the performance of machine learning solutions are significantly tied to data quality and volume [68]–[70]. In the case of optimization solvers, the regression networks of this study were trained on previously solved problem instances that were well representative of the distribution in which they will be deployed. A next milestone proof of principle for future research would be to successfully predict values for QAA at problem sizes larger than those trained on, unlocking the potential for the successful use of Amplitude Amplification at classically intractable scales.

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
