# OpenReview forum: "Amplitude Amplification for Quadratic Unconstrained Binary Optimization with Regression Based Neural Network Bootstrapping"
_purdue.edu/Purdue_University/PQAI/2025/Symposium — PQAI 2025 Oral_

### Official Review · Reviewer_Bfxn · 2025-07-23
**This work proposes a machine learning approach to predict critical free parameters in Quantum Amplitude Amplification (QAA) for solving linear QUBO problems.**

**Rating:** 6
**Confidence:** 4

**Review:**

This paper demonstrates that regression neural networks can accurately predict the critical parameter needed for successful Quantum Amplitude Amplification on QUBO problems, achieving sub-1% prediction errors and enabling practical QAA implementation.

Strengths:

The paper tackles a bottleneck in QAA implementation: determining the optimal parameter, which is essential for algorithm success but difficult to compute.

The results are impressive, particularly for Experiment 2 with multi-length QUBOs achieving μ = 0.04-0.05% average error and only 0.12% of predictions exceeding the 2% error threshold. This level of accuracy is practically meaningful for QAA success.

The progression from fixed-length to variable-length QUBOs demonstrates systematic investigation. The work addresses real quantum computing challenges and provides a concrete path toward hybrid classical-quantum optimization pipelines.

Weaknesses:

The evaluation is restricted to linear QUBOs (chain topology) with specific weight ranges [-200,200]. Real-world QUBO problems often have different topologies and weight distributions, limiting generalizability.

The largest QUBO length tested is 22. The authors acknowledge this limitation but don't provide compelling evidence that the approach will scale to quantum-advantageous problem sizes. Classical simulation constraints force this limitation, but it's a significant gap.

The paper lacks theoretical insight into why neural networks can capture the mathematical relationship between QUBO weights and optimal $p_s$ values.

The paper doesn't compare against existing approximation methods for $p_s$ determination mentioned in prior work [30], [32]. This makes it difficult to assess the relative merit of the ML approach.

Technical Issues:

The 2% error threshold is stated as necessary but not rigorously justified. More analysis of the relationship between prediction error and QAA success rates would strengthen the claims.

While LSTM-hybrid networks performed best, the paper doesn't provide strong theoretical rationale for why sequential processing of QUBO weights should be beneficial.

The models are only tested on similar distributions to training data. Cross-domain evaluation (different weight ranges, topologies) would better demonstrate robustness.

Minor Issues:
Some notation could be clearer (e.g., distinction between different uses of subscripts)

Figure 5 could benefit from error bars or confidence intervals

---

### Official Review · Reviewer_nt7H · 2025-07-25

**Rating:** 6
**Confidence:** 3

**Review:**

This paper studies the problem of applying classical ML methods (such as neural networks) to predict the free parameters for QAA in order to solve QUBO problems. Compared to the standard method of using SGD to optimize the parameters of the quantum circuit, this paper employs a neural network to predict them in a single shot. This advantage is that the prediction process is very fast and cheap. However, this method has several limitations. First, the effectiveness of this method largely depends on the training and generalization capabilities of the classical ML algorithm. For an industry-level optimization problem (where the quantum advantage is possible), obtaining good training data is a critical issue. We might only be able to use classical computers to find optimal or good parameters for small-scale instances. Then, to apply the model to larger problem instances, it becomes related to the out-of-distribution generalization problem, which is quite difficult in classical machine learning. Second, from a high-level perspective, I am not convinced that this approach is likely to demonstrate quantum advantage. Specifically, QAA or Grover's search could only provide a quadratic speedup. Then, for some NP-hard optimization problems, the complexity remains exponential. Moreover, the method in this paper lacks any theoretical guarantees. Nevertheless, the future work section proposes several interesting directions, and this paper could be strengthened if some of these directions were explored.

Overall, I think this paper could be accepted as a poster.

---

### Decision · Program_Chairs · 2025-07-29

Accept (Oral)